# Dual Space Gradient Descent for Online Learning

**Trung Le, Tu Dinh Nguyen, Vu Nguyen, Dinh Phung**
Centre for Pattern Recognition and Data Analytics
Deakin University, Australia
`{trung.l, tu.nguyen, v.nguyen, dinh.phung}@deakin.edu.au`

## Abstract

One crucial goal in kernel online learning is to bound the model size. Common approaches employ budget maintenance procedures to restrict the model sizes using removal, projection, or merging strategies. Although projection and merging, in the literature, are known to be the most effective strategies, they demand extensive computation whilst removal strategy fails to retain information of the removed vectors. An alternative way to address the model size problem is to apply random features to approximate the kernel function. This allows the model to be maintained directly in the random feature space, hence effectively resolve the curse of kernelization. However, this approach still suffers from a serious shortcoming as it needs to use a high dimensional random feature space to achieve a sufficiently accurate kernel approximation. Consequently, it leads to a significant increase in the computational cost. To address all of these aforementioned challenges, we present in this paper the *Dual Space Gradient Descent* (DualSGD), a novel framework that utilizes random features as an auxiliary space to maintain information from data points removed during budget maintenance. Consequently, our approach permits the budget to be maintained in a simple, direct and elegant way while simultaneously mitigating the impact of the dimensionality issue on learning performance. We further provide convergence analysis and extensively conduct experiments on five real-world datasets to demonstrate the predictive performance and scalability of our proposed method in comparison with the state-of-the-art baselines.

## 1 Introduction

Online learning represents a family of effective and scalable learning algorithms for incrementally building a predictive model from a sequence of data samples [1]. Unlike the conventional learning algorithms, which usually require a costly procedure to retrain the entire dataset when a new instance arrives [2], the goal of online learning is to utilize new incoming instances to improve the model given knowledge of the correct answers to previously processed data. The seminal line of work in online learning, referred to as *linear online learning* [3, 4], aims to learn a linear predictor in the input space. The key limitation of this approach lies in its oversimplified assumption in using a linear hyperplane to represent data that could possibly possess nonlinear dependency as commonly seen in many real-world applications. This inspires the work of *kernel online learning* [5, 6] that uses a linear model in the feature space to capture the nonlinearity of input data.

However, the kernel online learning approach suffers from the so-called *curse of kernelization* [7], that is, the model size linearly grows with the data size accumulated over time. A notable approach to address this issue is to use a budget [8, 9, 7, 10, 11]. The work in [7] leveraged the budgeted approach with stochastic gradient descent (SGD) [12, 13] wherein the learning procedure employed SGD and a budget maintenance procedure (e.g., removal, projection, or merging) was employed to maintain the model size. Although the projection and merging were shown to be effective [7], their associated computational costs render them impractical for large-scale datasets. An alternative way to address the curse of kernelization is to use random features [14] to approximate a kernel function

**Acknowledgment**: This work is partially supported by the Australian Research Council under the Discovery Project DP160109394.

[15, 16]. The work in [16] proposed to transform data from the input space to the random-feature space, and then performed SGD in the feature space. However, in order for this approach to achieve good kernel approximation, excessive number of random features is required, hence could lead to serious computational issue.

In this paper, we propose the *Dual Space Gradient Descent* (DualSGD) to address the computational problem encountered in the projection and merging strategies in the budgeted approach [8, 9, 17, 7] and the excessive number of random features in the random feature approach [15, 16]. In particular, the proposed DualSGD utilizes the random-feature space as an auxiliary space to store the information of the vectors that have been discarded during the budget maintenance process. More specifically, the DualSGD uses a *provision vector* in the random-feature space to store the information of all vectors being removed. This allows us to propose a novel budget maintenance strategy, named *k-merging*, which unifies the removal, projection, and merging strategies.

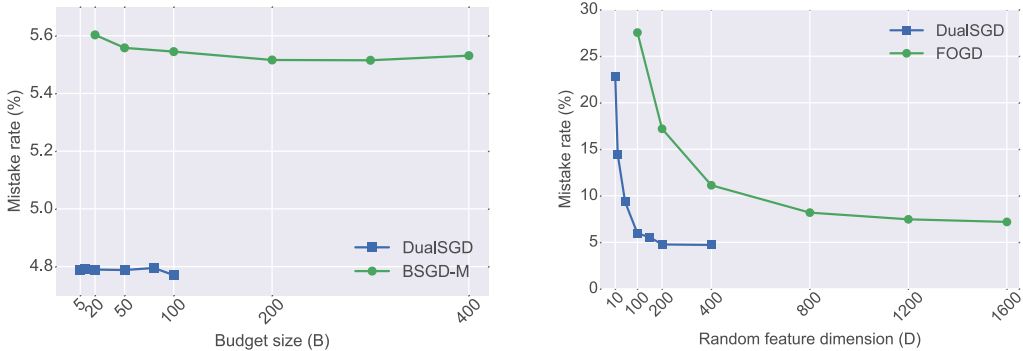

Figure 1: Comparison of DualSGD with BSGD-M and FOGD on the cod-rna dataset. Left: DualSGD vs. BSGD-M when $B$ is varied. Right: DualSGD vs. FOGD when $D$ is varied.

Our proposed DualSGD advances the existing works in the budgeted and random-feature approaches in twofold. Firstly, since the goal of using random features is to approximate the original feature space as much as possible, the proposed *k-merging* of DualSGD can preserve the information of the removed vectors more effectively than the existing budget maintenance strategies. For example comparing with the budgeted SGD using merging strategy (BSGD-M) [7], as shown in Fig. 1 (left), the DualSGD with a small budget size ($B = 5$) can gain a significant better mistake rate than that of BSGD-M with a 80-fold larger budget size ($B = 400$). Secondly, since the core part of the model (i.e., the vectors in the support set) is stored in the feature space and the auxiliary part (i.e., the removed vectors) is stored in the random-feature space, our DualSGD can significantly reduce the influence of the number of random features to the learning performance. For example comparing with the Fourier Online Gradient Descent (FOGD) [16], as shown in Fig. 1 (right), the DualSGD with a small number of random features ($D = 20$) can achieve a comparable mistake rate to that of FOGD with a 40-fold larger number of random features ($D = 800$) and the DualSGD with a medium value of number of random features ($D = 100$) achieves a predictive performance that would not be reached by FOGD (the detail of comparison in computational complexities of our DualSGD and FOGD can be found in Section 3 in the supplementary material).

To provide theoretical foundation for DualSGD, we develop an extensive convergence analysis for a wide spectrum of loss functions including Hinge, Logistic, and smooth Hinge [18] for classification task and $\ell_1$, $\varepsilon$-insensitive for regression. We conduct extensive experiments on five real-world datasets to compare the proposed method with the state-of-the-art online learning methods. The experimental results show that our proposed DualSGD achieves the most optimal predictive results in almost all cases, whilst its execution time is much faster than the baselines.

## 2 Dual Space Gradient Descent for Online Learning

### 2.1 Problem Setting

We propose to solve the following optimization problem: $\min_{\mathbf{w}} \mathcal{J}(\mathbf{w})$ whose objective function is defined for online setting as follows:

$$\mathcal{J}(\mathbf{w}) \equiv \frac{\lambda}{2} \|\mathbf{w}\|^2 + \mathbb{E}_{(\boldsymbol{x},y) \sim p_{\mathcal{X},\mathcal{Y}}} [l(\mathbf{w}, \boldsymbol{x}, y)] \tag{1}$$

where $\boldsymbol{x} \in \mathbb{R}^M$ is the data vector, $y$ the label, $p_{\mathcal{X},\mathcal{Y}}$ denotes the joint distribution over $\mathcal{X} \times \mathcal{Y}$ with the data domain $\mathcal{X}$ and label domain $\mathcal{Y}$, $l(\mathbf{w}, \boldsymbol{x}, y)$ is a convex loss function with parameters $\mathbf{w}$, and $\lambda \geq 0$ is a regularization parameter. A kernelization of the loss function introduces a nonlinear function $\Phi$ that maps $\boldsymbol{x}$ from the input space to a feature space. A classic example is the Hinge loss: $l(\mathbf{w}, \boldsymbol{x}, y) = \max\left(0, 1 - y\mathbf{w}^\top \Phi(\boldsymbol{x})\right)$.

## 2.2 The Key Ideas of the Proposed DualSGD

Our key motivations come from the shortcomings of three current budget maintenance strategies: removal, projection and merging. The removal strategy fails to retain information of the removed vectors. Although the projection strategy can overcome this problem, it requires a costly procedure to compute the inverse of an $B \times B$ matrix wherein $B$ is the budget size, typically in the cubic complexity of $B$. On the other hand, the merging strategy needs to estimate the preimage of a vector in the feature space, leading to a significant information loss and requiring extensive computation. Our aim is to find an approach to simultaneously retain the information of the removed vectors accurately, and perform budget maintenance efficiently.

To this end, we introduce the *k-merging*, a new budget maintenance approach that unifies three aforementioned budget maintenance strategies under the following interpretation. For $k = 1$, the proposed $k$-merging can be seen as a hybrid strategy of removal and projection. For $k = 2$, it can be regarded as the standard merging. Moreover, our proposed $k$-merging strategy enables an arbitrary number of vectors to be conveniently merged. Technically, we employ a vector in the random-feature space [14], called *provision vector* $\tilde{\mathbf{w}}$, to retain the information of all removed vectors. When $k$-merging is invoked, the most redundant $k$ vectors are sorted out, e.g., $\boldsymbol{x}_{i_1}, \ldots, \boldsymbol{x}_{i_k}$ and we increment $\tilde{\mathbf{w}}$ as $\tilde{\mathbf{w}} = \tilde{\mathbf{w}} + \sum_{j=1}^k \alpha_{i_j} \boldsymbol{z}\left(\boldsymbol{x}_{i_j}\right)$ where $\alpha_{i_j}$ is the coefficient of support vector associated with $\boldsymbol{x}_{i_j}$, and $\boldsymbol{z}\left(\boldsymbol{x}_{i_j}\right)$ denotes the mapping function from the input space to the random feature space. The advantage of using the random-feature space as an auxiliary space is twofold: 1) the information loss is negligible since the random-feature space is designed to approximate the original feature space, and 2) the operations in budget maintenance strategy are direct and economic.

---

**Algorithm 1** The learning of Dual Space Gradient Descent.

---

   **Input:** Kernel $K$, regularization parameter $\lambda$, budget $B$, random feature dimension $D$.
1:   $\hat{\mathbf{w}}_1 = \mathbf{0}$;   $\tilde{\mathbf{w}}_1 = \mathbf{0}$;   $b = 0$;   $I_0 = \emptyset$
2: **for** $t = 1, \ldots, T$ **do**
3:     $(\boldsymbol{x}_t, y_t) \sim p_{\mathcal{X},\mathcal{Y}}$
4:     $\hat{\mathbf{w}}_{t+1} = \frac{t-1}{t}\hat{\mathbf{w}}_t$;   $\tilde{\mathbf{w}}_{t+1} = \frac{t-1}{t}\tilde{\mathbf{w}}_t$
5:     **if** $\nabla_o l\left(y_t, o_t^h\right) \neq 0$ **then**
6:        $I_t = I_{t-1} \cup \{t\}$
7:        $\hat{\mathbf{w}}_{t+1} = \hat{\mathbf{w}}_{t+1} - \frac{1}{\lambda t}\nabla_o l\left(y_t, o_t^h\right)\Phi(\boldsymbol{x}_t)$
8:        **if** $|I_t| > B$ **then**
9:           invokes $k$-merging$(I_t, \hat{\mathbf{w}}_{t+1}, \tilde{\mathbf{w}}_{t+1})$
10:       **end if**
11:     **end if**
12: **end for**
   **Output:** $\mathbf{w}_{T+1}^h = \hat{\mathbf{w}}_{T+1} \oplus \tilde{\mathbf{w}}_{T+1}$ .

---

## 2.3 The Proposed Algorithm

In our proposed DualSGD, the model is distributed into two spaces: the feature and random-feature spaces with a *hybrid* vector $\mathbf{w}_t^h$ defined as: $\mathbf{w}_t^h \triangleq \hat{\mathbf{w}}_t \oplus \tilde{\mathbf{w}}_t$. Here we note that the kernel part $\hat{\mathbf{w}}_t$ and the provision part $\tilde{\mathbf{w}}_t$ lie in two different spaces, thus for convenience we define an abstract operator $\oplus$ to allow the addition between them, which implies that the decision function crucially depends on both kernel and provision parts

$$\left\langle \mathbf{w}_t^h, \boldsymbol{x} \right\rangle \triangleq \left\langle (\hat{\mathbf{w}}_t \oplus \tilde{\mathbf{w}}_t), \boldsymbol{x} \right\rangle \triangleq \hat{\mathbf{w}}_t^\top \Phi(\boldsymbol{x}) + \tilde{\mathbf{w}}_t^\top \boldsymbol{z}(\boldsymbol{x})$$

We employ one vector $\tilde{\mathbf{w}}_t$ in the random-feature space to preserve the information of the discarded vectors, that are outside $I_t$ – the set of indices of all support vectors in $\hat{\mathbf{w}}_t$. When an instance arrives and the model size exceeds the budget $B$, the budget maintenance procedure $k$-merging$(I_t, \hat{\mathbf{w}}_{t+1}, \tilde{\mathbf{w}}_{t+1})$ is invoked to adjust $\hat{\mathbf{w}}_{t+1}$ and $\tilde{\mathbf{w}}_{t+1}$, accordingly. Our proposed DualSGD is summarized in Algorithm 1 where we note that, $l(y, o)$ is another representation of convex loss function w.r.t the variable

$o$ (e.g., the Hinge loss given by $l(y,o) = \max(0, 1 - yo)$), and $o_t^h = \hat{\mathbf{w}}_t^\top \Phi(\boldsymbol{x}) + \tilde{\mathbf{w}}_t^\top \boldsymbol{z}(\boldsymbol{x})$ (i.e., hybrid objective value).

## 2.4  $k$-merging Budget Maintenance Strategy

Crucial to our proposed DualSGD in Algorithm 1 is the k-merging routine to allow efficient merging of k arbitrary vectors. We summarize the key steps for k-merging in Algorithm 2. In particular, we first select $k$ support vectors whose corresponding coefficients $(\alpha_{i_1}, \alpha_{i_2}, ..., \alpha_{i_k})$ have the smallest absolute values (cf. line 1). We then approximate them by $\boldsymbol{z}(\boldsymbol{x}_{i_1}), \ldots, \boldsymbol{z}(\boldsymbol{x}_{i_k})$ and merge them by updating the provision vector as $\tilde{\mathbf{w}}_{t+1} = \tilde{\mathbf{w}}_{t+1} + \sum_{j=1}^{k} \alpha_{i_j} \boldsymbol{z}(\boldsymbol{x}_{i_j})$ (cf. line 2). Finally, we remove the chosen vectors from the kernel part $\hat{\mathbf{w}}_{t+1}$ (cf. line 2).

## 2.5  Convergence Analysis

In this section, we present the convergence analysis for our proposed algorithm. We first prove that with a high probability $f_t^h(\boldsymbol{x})$ (i.e., hybrid decision function and cf. 3) is a good approximation of $f_t(\boldsymbol{x})$ for all $\boldsymbol{x}$ and $t$ (cf. Theorem 1). Let $\mathbf{w}^\star$ be the optimal solution of the optimization problem defined in Eq. (1): $\mathbf{w}^\star = \underset{\mathbf{w}}{\operatorname{argmin}} \mathcal{J}(\mathbf{w})$. We then prove that if $\{\mathbf{w}_t\}_{t=1}^\infty$ is constructed as in Eq. (2), this sequence rapidly converges to $\mathbf{w}^\star$ or $f_t(\boldsymbol{x}) = \mathbf{w}_t^\top \Phi(\boldsymbol{x})$ rapidly approaches the optimal decision function (cf. Theorems 2, 3). Therefore, the decision function $f_t^h(\boldsymbol{x})$ also rapidly approaches the optimal decision function. Our analysis can be generalized for the general $k$-merging strategy, but for comprehensibility we present the analysis for the 1-merging case (i.e., $k = 1$).

We assume that the loss function used in the analysis satisfies the condition $|\nabla_o l(y, o)| \leq A$, $\forall y, o$, where $A$ is a positive constant. A wide spectrum of loss functions including Hinge, logistic, smooth Hinge [18], $\ell_1$, and $\varepsilon$-insensitive satisfy this condition and hence are appropriate for this convergence analysis. We further assume that $\|\Phi(\boldsymbol{x})\| = K(\boldsymbol{x}, \boldsymbol{x})^{1/2} = 1$, $\forall \boldsymbol{x}$. Let $\beta_t$ be a binary random variable which indicates whether the budget maintenance procedure is performed at the iteration $t$ (i.e., the event $\nabla_o l(y_t, o_t^h) \neq 0$). We assume that if $\beta_t = 1$, the vector $\Phi(\boldsymbol{x}_{i_t})$ is selected to move to the random-feature space. Without loss of generality, we assume that $i_t = t$ since we can arrange the data instances so as to realize it. We define

$$g_t^h = \lambda \mathbf{w}_t + \nabla_o l(y_t, f_t^h(\boldsymbol{x}_t)) \Phi(\boldsymbol{x}_t) \quad \text{and} \quad \mathbf{w}_{t+1} = \mathbf{w}_t - \eta_t g_t^h \tag{2}$$

$$f_t(\boldsymbol{x}) = \mathbf{w}_t^\top \Phi(\boldsymbol{x}) = \sum_{j=1}^{t} \alpha_j K(\boldsymbol{x}_j, \boldsymbol{x})$$

$$f_t^h(\boldsymbol{x}) = \hat{\mathbf{w}}_t^\top \Phi(\boldsymbol{x}_t) + \tilde{\mathbf{w}}_t^\top \boldsymbol{z}(\boldsymbol{x}_t) = \sum_{j=1}^{t} \alpha_j (1 - \beta_j) K(\boldsymbol{x}_j, \boldsymbol{x}) + \sum_{j=1}^{t} \alpha_j \beta_j \tilde{K}(\boldsymbol{x}_j, \boldsymbol{x}) \tag{3}$$

where $\tilde{K}(\boldsymbol{x}, \boldsymbol{x}') = \boldsymbol{z}(\boldsymbol{x})^\top \boldsymbol{z}(\boldsymbol{x}')$ is the approximated kernel induced by the random-feature space, and the learning rate $\eta_t = \frac{1}{\lambda t}$.

Theorem 1 establishes that $f_t^h(.)$ is a good approximation of $f_t(\boldsymbol{x})$ with a high probability, followed by Theorem 2 which establishes the bound on the regret.

---

**Algorithm 2** $k$-merging Budget Maintenance Procedure.

    **procedure** $k$-merging$(I_t, \hat{\mathbf{w}}_{t+1}, \tilde{\mathbf{w}}_{t+1})$
    // Assume that $\hat{\mathbf{w}}_{t+1} = \sum_{j \in I_t} \alpha_j \Phi(\boldsymbol{x}_j)$
1: $(i_1, \ldots, i_k) = $k-$\underset{j \in I_t}{\operatorname{argmin}} |\alpha_j|$;    $I_t = I_t \setminus \{i_1, \ldots, i_k\}$
2: $\tilde{\mathbf{w}}_{t+1} = \tilde{\mathbf{w}}_{t+1} + \sum_{j=1}^{k} \alpha_{i_j} \boldsymbol{z}(\boldsymbol{x}_{i_j})$;    $\hat{\mathbf{w}}_{t+1} = \hat{\mathbf{w}}_{t+1} - \sum_{j=1}^{k} \alpha_{i_j} \Phi(\boldsymbol{x}_{i_j})$
    **endprod**

---

**Theorem 1.** *With a probability at least* $1 - \theta = 1 - 2^8 \left( \frac{\sigma_\mu A d_{\mathcal{X}}}{\lambda \varepsilon} \right) \exp\left( -\frac{D \lambda^2 \varepsilon^2}{4(M+2) A^2} \right)$ *where $M$ is the dimension of input space, $D$ is the dimension of random feature space, $d_{\mathcal{X}}$ denotes the diameter of the compact set $\mathcal{X}$, and the constant $\sigma_\mu$ is defined as in [14], we have*

*i)* $\left| f_t(\boldsymbol{x}) - f_t^h(\boldsymbol{x}) \right| \leq \varepsilon$ *for all $t > 0$ and $\boldsymbol{x} \in \mathcal{X}$.*

*ii)* $\mathbb{E}\left[ \left| f_t(\boldsymbol{x}) - f_t^h(\boldsymbol{x}) \right| \right] \leq A^{-1} \lambda \varepsilon \sum_{j=1}^{t} \mathbb{E}\left[ \alpha_j^2 \right]^{1/2} \mu_j^{1/2}$ *where $\mu_j = p(\beta_j = 1)$.*

Theorem 1 shows that with a high probability $f_t^h(\boldsymbol{x})$ can approximate $f_t(\boldsymbol{x})$ with an $\varepsilon$-precision. It also indicates that to decrease the gap $\left| f_t(\boldsymbol{x}) - f_t^h(\boldsymbol{x}) \right|$, when performing budget maintenance, we should choose the vectors whose coefficients have smallest absolute values to move to the random-feature space.

**Theorem 2.** *The following statement guarantees for all $T$*

$$\mathbb{E}\left[\mathcal{J}\left(\overline{\mathbf{w}}_T\right)\right] - \mathcal{J}\left(\mathbf{w}^\star\right) \leq \mathbb{E}\left[\frac{1}{T}\sum_{t=1}^T \mathcal{J}\left(\mathbf{w}_t\right) - \mathcal{J}\left(\mathbf{w}^\star\right)\right] \leq \frac{8A^2\left(\log T + 1\right)}{\lambda T} + \frac{1}{T}W\sum_{t=1}^T \mathbb{E}\left[M_t^2\right]^{1/2}$$

*where $\overline{\mathbf{w}}_T = \frac{1}{T}\sum_{t=1}^T \mathbf{w}_t$, $M_t = \nabla_o l\left(y_t, f_t(\boldsymbol{x}_t)\right) - \nabla_o l\left(y_t, f_t^h(\boldsymbol{x}_t)\right)$, and $W = 2A\left(1 + \sqrt{5}\right)\lambda^{-1}$.*

If a smooth loss function is used, we can quantify the gap in more detail and with a high probability, the gap is negligible and this is shown in Theorem 3.

**Theorem 3.** *Assume that $l(y, o)$ is a $\gamma$-strongly smooth loss function. With a probability at least $1 - 2^8\left(\frac{\sigma_\mu A d_\mathcal{X}}{\lambda \varepsilon}\right)\exp\left(-\frac{D\lambda^2\varepsilon^2}{4(M+2)A^2}\right)$, we have*

$$\mathbb{E}\left[\mathcal{J}\left(\overline{\mathbf{w}}_T\right)\right] - \mathcal{J}\left(\mathbf{w}^\star\right) \leq \mathbb{E}\left[\frac{1}{T}\sum_{t=1}^T \mathcal{J}\left(\mathbf{w}_t\right) - \mathcal{J}\left(\mathbf{w}^\star\right)\right] \leq \frac{8A^2\left(\log T + 1\right)}{\lambda T} + \frac{1}{T}W\gamma\varepsilon\sum_{t=1}^T\left(\frac{\sum_{j=1}^t \mu_j}{t}\right)^{1/2}$$

$$\leq \frac{8A^2\left(\log T + 1\right)}{\lambda T} + W\gamma\varepsilon$$

## 3 Experiments

In this section, we conduct comprehensive experiments to quantitatively evaluate the performance of our proposed Dual Space Gradient Descent (DualSGD) on binary classification, multiclass classification and regression tasks under online settings. Our main goal is to examine the scalability, classification and regression capabilities of DualSGDs by directly comparing them with those of several recent state-of-the-art online learning approaches using a number of real-world datasets with a wide range of sizes. In what follows, we present the data statistics, experimental setup, results and our observations.

### 3.1 Data Statistics and Experimental Setup

We use 5 datasets which are *ijcnn1*, *cod-rna*, *poker*, *year*, and *airlines*. The datasets where purposely are selected with various sizes in order to clearly expose the differences among scalable capabilities of the models. Three of which are large-scale datasets with hundreds of thousands and millions of data points (*year*: $515,345$; *poker*: $1,025,010$; and *airlines*: $5,929,413$), whilst the rest are medium size databases (*ijcnn1*: $141,691$ and *cod-rna*: $331,152$). These datasets can be downloaded from LIBSVM[1] and UCI[2] websites, except the *airlines* which was obtained from American Statistical Association (ASA[3]). For the *airlines* dataset, our aim is to predict whether a flight will be delayed or not under binary classification setting, and how long (in minutes) the flight will be delayed in terms of departure time under regression setting. A flight is considered *delayed* if its delay time is above 15 minutes, and *non-delayed* otherwise. Following the procedure in [19], we extract 8 features for flights in the year of 2008, and then normalize them into the range [0,1].

For each dataset, we perform 10 runs on each algorithm with different random permutations of the training data samples. In each run, the model is trained in a single pass through the data. Its prediction result and time spent are then reported by taking the average together with the standard deviation over all runs. For comparison, we employ 11 state-of-the-art online kernel learning methods: perceptron [5], online gradient descent (OGD) [6], randomized budget perceptron (RBP) [9], forgetron [8] projectron, projectron++ [20], budgeted passive-aggressive simple (BPAS) [17], budgeted SGD using merging strategy (BSGD-M) [7], bounded OGD (BOGD) [21], Fourier OGD (FOGD) and Nystrom OGD (NOGD) [16]. Their implementations are published as a part of LIBSVM, BudgetedSVM[4] and LSOKL[5] toolboxes. We use a Windows machine with 3.46GHz Xeon processor and 96GB RAM to conduct our experiments.

## 3.2 Model Evaluation on the Effect of Hyperparameters

In the first experiment, we investigate the effect of hyperparameters, i.e., budget size $B$, merging size $k$ and random feature dimension $D$ (cf. Section 2) on the performance behavior of DualSGD. Particularly, we conduct an initial analysis to quantitatively evaluate the sensitivity of these hyperparameters and their impact on the predictive accuracy and wall-clock time. This analysis provides an approach to find the best setting of hyperparameters. Here the DualSGD with Hinge loss is trained on the *cod-rna* dataset under the online classification setting.

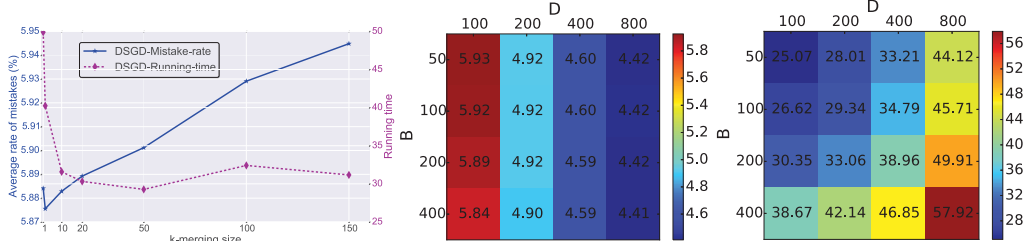

Figure 2: The effect of $k$-merging size on the mistake rate and running time (left). The effect of budget size $B$ and random feature dimension $D$ on the mistake rate (middle) and running time (right).

First we set $B = 200$, $D = 100$, and vary $k$ in the range of $1, 2, 10, 20, 50, 100, 150$. For each setting, we run our models and record the average mistake rates and running time as shown in Fig. 2 (left). There is a pattern that the classification error increases for larger $k$ whilst the wall-clock time decreases. This represents the trade-off between model discriminative performance and model computational complexity via the number of merging vectors. In this analysis, we can choose $k = 20$ to balance the performance and computational cost.

Fixing $k = 20$, we vary $B$ and $D$ in 4 values doubly increasing from 50 to 400 and from 100 to 800, respectively, to evaluate the prediction performance and execution time. Fig. 2 depicts the average mistake rates (middle) and running time in seconds (right) as a heat map of these values. These visualizations indicate that the higher $B$ and $D$ produce better classification results, but hurt the training speed of the model. We found that increasing the dimension of random feature space from 100 to 800 at $B = 50$ significantly reduces the mistake rates by $25\%$, at the same time increases the wall-clock time by $76\%$. The same pattern with less effect is observed when increasing the budget size $B$ from 50 to 400 at $D = 100$ (mistake rate decreases by $1.5\%$, time increases by $54\%$). For a good trade-off between classification performance and computational cost, we select $B = 100$ and $D = 200$ which achieves fairly comparable classification result and running time.

## 3.3 Online Classification

We now examine the performances of DualSGDs in the online classification task. We use four datasets: *cod-rna*, *ijcnn1, poker* and *airlines* (delayed and non-delayed labels). We create two versions of our approach: DualSGD with Hinge loss (DualSGD-Hinge) and DualSGD with Logistic loss (DualSGD-Logit). It is worth mentioning that the Hinge loss is not a smooth function with undefined gradient at the point that the classification confidence $yf(x) = 1$. Following the subgradient definition, in our experiment, we compute the gradient given the condition that $yf(x) < 1$, and set it to $0$ otherwise.

**Hyperparameters setting.** There are a number of different hyperparameters for all methods. Each method requires a different set of hyperparameters, e.g., the regularization parameters ($\lambda$ in DualSGD), the learning rates ($\eta$ in FOGD and NOGD), and the RBF kernel width ($\gamma$ in all methods). Thus, for a fair comparison, these hyperparameters are specified using cross-validation on a subset of data.

In particular, we further partition the training set into $80\%$ for learning and $20\%$ for validation. For large-scale databases, we use only $1\%$ of dataset, so that the searching can finish within an acceptable time budget. The hyperparameters are varied in certain ranges and selected for the best performance on the validation set. The ranges are given as follows: $C \in \{2^{-5}, 2^{-3}, ..., 2^{15}\}$, $\lambda \in \{2^{-4}/N, 2^{-2}/N, ..., 2^{16}/N\}$, $\gamma \in \{2^{-8}, 2^{-4}, 2^{-2}, 2^{0}, 2^{2}, 2^{4}, 2^{8}\}$, and $\eta \in \{2^{-4}, 2^{-3}, ..., 2^{-1}, 2^{1}, 2^{2} ..., 2^{4}\}$ where $N$ is the number of data points. The budget size $B$, merging size $k$ and random feature dimension $D$ of DualSGD are selected following the approach described in Section 3.2. For the budget size $\hat{B}$ in NOGD and Pegasos algorithm, and the feature dimension $\hat{D}$ in FOGD for each dataset, we use identical values to those used in Section 7.1.1 of [16].

Table 1: Mistake rate (%) and execution time (seconds). The notation $[k; B; D; \hat{B}; \hat{D}]$ denotes the merging size $k$, the budget sizes $B$ and $\hat{B}$ of DualSGD-based models and other budgeted algorithms, and the number of random features $D$ and $\hat{D}$ of DualSGD and FOGD, respectively.

| Dataset $\left[k \mid B \mid D \mid \hat{B} \mid \hat{D}\right]$ | cod-rna [20 \| 100 \| 200 \| 400 \| 1,600] | | ijcnn1 [20 \| 100 \| 200 \| 1,000 \| 4,000] | |
|---|---|---|---|---|
| **Algorithm** | **Mistake Rate** | **Time** | **Mistake Rate** | **Time** |
| Perceptron | 9.79±0.04 | 1,393.56 | 12.85±0.09 | 727.90 |
| OGD | 7.81±0.03 | 2,804.01 | 10.39±0.06 | 960.44 |
| RBP | 26.02±0.39 | 85.84 | 15.54±0.21 | 54.29 |
| Forgetron | 28.56±2.22 | 102.64 | 16.17±0.26 | 60.54 |
| Projectron | 11.16±3.61 | 97.38 | 12.98±0.23 | 59.37 |
| Projectron++ | 17.97±15.60 | 1,799.93 | 9.97±0.09 | 749.70 |
| BPAS | 11.97±0.09 | 92.08 | 10.68±0.05 | 55.44 |
| BSGD-M | 5.33±0.04 | 184.58 | 9.14±0.18 | 1,562.61 |
| BOGD | 38.13±0.11 | 104.60 | 10.87±0.18 | 55.99 |
| FOGD | 7.15±0.03 | 53.45 | 9.41±0.03 | 25.93 |
| NOGD | 7.83±0.06 | 105.18 | 10.43±0.08 | 59.36 |
| DualSGD-Hinge | 4.92±0.25 | **28.29** | **8.35±0.20** | **12.12** |
| DualSGD-Logit | **4.83±0.21** | 31.96 | 8.82±0.24 | 13.30 |

| Dataset $[S]$ $\left[k \mid B \mid D \mid \hat{B} \mid \hat{D}\right]$ | poker [20 \| 100 \| 200 \| 1,000 \| 4,000] | | airlines [20 \| 100 \| 200 \| 1,000 \| 4,000] | |
|---|---|---|---|---|
| **Algorithm** | **Mistake Rate** | **Time** | **Mistake Rate** | **Time** |
| FOGD | 52.28±0.04 | 928.89 | 20.98±0.01 | 1,270.75 |
| NOGD | **44.90±0.16** | 4,920.33 | 25.56±0.01 | 3,553.50 |
| DualSGD-Hinge | 46.73±0.22 | 139.87 | **19.28±0.00** | **472.21** |
| DualSGD-Logit | 46.65±0.14 | **133.50** | **19.28±0.00** | 523.23 |

**Results.** Table 1 reports the average classification results and execution time after the methods see all data samples. Note that for two biggest datasets (*poker*, *airlines*) that consist of millions of data points, we only include the fast algorithms FOGD, NOGD and DualSGDs. The other methods would exceed the time limit, which we set to two hours, when running on such data as they suffer from serious computation issue. From these results, we can draw key observations below.

The budgeted online approaches show their effectiveness with substantially faster computation than the ones without budgets. More specifically, the execution time of our proposed models is several orders of magnitude (100 times) lower than that of regular online algorithms (e.g., 28.29 seconds compared with 2,804 seconds for *cod-rna* dataset). Moreover, our models are twice as fast as the recent fast algorithm FOGD for *cod-rna* and *ijcnn1* datasets, and approximately eight and three times for vast-sized data *poker* and *airlines*. This is because the DualSGDs maintain a sparse budget of support vectors and a low random feature space, whose size and dimensionality are 10 times and 20 times smaller than those of other methods.

Second, in terms of classification, the DualSGD-Hinge and DualSGD-Logit outperform other methods for almost all datasets except the *poker* data. In particular, the DualSGD-based methods achieve the best mistake rates 4.83±0.21, 8.35±0.20, 19.28±0.00 for the *cod-rna*, *ijcnn1* and *airlines* data, that are, respectively, 32.4%, 11.3%, 8.8% lower than the error rates of the second best models – two recent approaches FOGD and NOGD. For *poker* dataset, our methods obtain fairly comparable results with that of the NOGD, but still surpass the FOGD with a large margin. The reason is that the DualSGD uses a dual space: a kernel space containing core support vectors and a random feature space keeping the projections of the core vectors that are removed from the budget in kernel space. This would minimize the information loss when the model performs budget maintenance.

Finally, two versions of DualSGDs demonstrate similar discriminative performances and computational complexities wherein the DualSGD-Logit is slightly slower due to the additional exponential operators. All of these observations validate the effectiveness and efficiency of our proposed technique. Thus, we believe that our approximation machine is a promising technique for building scalable online kernel learning algorithms for large-scale classification tasks.

### 3.4 Online Regression

The last experiment addresses the online regression problem to evaluate the capabilities of our approach with two proposed loss functions: $\ell_1$ and $\varepsilon$-insensitive losses. Incorporating these loss functions creates two versions: DualSGD-$\varepsilon$, DualSGD-$\ell_1$. We use two datasets: *year* and *airlines* (delay minutes), and six baselines: RBP, Forgetron, Projectron, BOGD, FOGD and NOGD.

Table 2: Root mean squared error (RMSE) and execution time (seconds) of 6 baselines and 2 versions of our DualSGDs. The notation $[k; B; D; \hat{B}; \hat{D}]$ denotes the same meaning as those in Table 1.

| Dataset | year | | airlines | |
|---|---|---|---|---|
| $\left[k \mid B \mid D \mid \hat{B} \mid \hat{D}\right]$ | $[20 \mid 100 \mid 200 \mid 400 \mid 1,600]$ | | $[20 \mid 100 \mid 200 \mid 1,000 \mid 2,000]$ | |
| Algorithm | RMSE | Time | RMSE | Time |
| RBP | 0.19±0.00 | 605.42 | 36.51±0.00 | 3,418.89 |
| Forgetron | 0.19±0.00 | 904.09 | 36.51±0.00 | 5,774.47 |
| Projectron | 0.14±0.00 | 605.19 | 36.14±0.00 | 3,834.19 |
| BOGD | 0.20±0.00 | 596.10 | 35.73±0.00 | 3,058.96 |
| FOGD | 0.16±0.00 | 76.70 | 53.16±0.01 | 646.15 |
| NOGD | 0.14±0.00 | 607.37 | **34.74±0.00** | 3,324.38 |
| DualSGD-$\varepsilon$ | 0.13±0.00 | 48.01 | 36.20±0.01 | 457.30 |
| DualSGD-$\ell_1$ | **0.12±0.00** | **47.29** | 36.20±0.01 | **443.39** |

**Hyperparameters setting.** We adopt the same hyperparameter searching procedure for online classification task as in Section 3.3. Furthermore, for the budget size $\hat{B}$ and the feature dimension $\hat{D}$ in FOGD, we follow the same strategy used in Section 7.1.1 of [16]. More specifically, these hyperparameters are separately set for different datasets as reported in Table 2. They are chosen such that they are roughly proportional to the number of support vectors produced by the batch SVM algorithm in LIBSVM running on a small subset. The aim is to achieve competitive accuracy using a relatively larger budget size for tackling more challenging regression tasks.

**Results.** Table 2 reports the average regression errors and computation costs after the methods see all data samples. From these results, we can draw some observations below.

Our proposed models enjoy a significant advantage in computational efficacy whilst achieve better (for *year* dataset) or competitive regression results (for *airlines* dataset) with other methods. The DualSGD, again, secures the best performance in terms of model sparsity. Among the baselines, the FOGD is the fastest, that is, its time costs can be considered to compare with those of our methods, but its regression performances are worse. The remaining algorithms usually obtain better results, but is paid by the sacrifice of scalability.

Finally, comparing the capability of two DualSGD's variants, both models demonstrate similar regression capabilities and computational complexities wherein the DualSGD-$\ell_1$ is slightly faster due to its simpler operator in computing the gradient. Besides, its regression scores are also lower or equal to those of DualSGD-$\varepsilon$. These observations, once again, verifies the effectiveness and efficiency of our proposed techniques. Therefore the DualSGD is also a promising machine to perform online regression task for large-scale datasets.

## 4 Conclusion

In this paper, we have proposed Dual Space Gradient Descent (DualSGD) that overcomes the computational problem in the projection and merging strategies in Budgeted SGD (BSGD) and the excessive number of random features in Fourier Online Gradient Descent (FOGD). More specifically, we have employed the random features to form an auxiliary space for storing the vectors being removed during the budget maintenance process. This makes the operations in budget maintenance simple and convenient. We have further presented the convergence analysis that is appropriate for a wide spectrum of loss functions. Finally, we have conducted the extensive experiments on several benchmark datasets to prove the efficiency and accuracy of the proposed method.

## Footnotes

[1]https://www.csie.ntu.edu.tw/~cjlin/libsvmtools/datasets/

[2]https://archive.ics.uci.edu/ml/datasets.html

[3]http://stat-computing.org/dataexpo/2009/.

[4]http://www.dabi.temple.edu/budgetedsvm/index.html

[5]http://lsokl.stevenhoi.com/

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
