[Supplementary Material · supp_DSGD.pdf]

# Supplementary Material for Dual Space Gradient Descent for Online Learning

**Trung Le, Tu Dinh Nguyen, Vu Nguyen, Dinh Phung**
Centre for Pattern Recognition and Data Analytics
Deakin University, Australia
{trung.l, tu.nguyen, v.nguyen, dinh.phung}@deakin.edu.au

## 1  Suitability of Loss Functions

In this section, we present the suitability of the loss functions for Hinge, smooth Hinge, and Logistic for classification and $\ell_1$, and $\varepsilon$-insensitive for regression. We prove that these losses satisfy the condition: there exists a positive constant $A$ such that $|\nabla_o l(y, o)| \leq A$, $\forall y, o$. For each loss, we show its two forms used in the paper w.r.t $o$ and $\mathbf{w}$.

**Hinge loss**

$$l(y, o) = \max(0, 1 - yo)$$
$$l(\mathbf{w}, \boldsymbol{x}, y) = \max\left(0, 1 - y\mathbf{w}^\top \Phi(\boldsymbol{x})\right)$$
$$\nabla_o l(y, o) = -\mathbb{I}_{yo \leq 1} y$$
$$|\nabla_o l(y, o)| = |\mathbb{I}_{yo \leq 1}| \leq 1 = A$$

**Logistic loss**

$$l(y, o) = \log\left(1 + e^{-yo}\right)$$
$$l(\mathbf{w}, \boldsymbol{x}, y) = \log\left(1 + e^{-y\mathbf{w}^\top \Phi(\boldsymbol{x})}\right)$$
$$\nabla_o l(y, o) = \frac{-ye^{-yo}}{e^{-yo} + 1}$$
$$|\nabla_o l(y, o)| = \left|\frac{e^{-yo}}{e^{-yo} + 1}\right| < 1 = A$$

**Smooth Hinge loss [4]**

$$l(y, o) = \begin{cases} 0 & \text{if } yo > 1 \\ 1 - yo - \frac{\tau}{2} & \text{if } yo < 1 - \tau \\ \frac{1}{2\tau}(1 - yo)^2 & \text{otherwise} \end{cases}$$

$$l(\mathbf{w}, \boldsymbol{x}, y) = \begin{cases} 0 & \text{if } y\mathbf{w}^\top \Phi(\boldsymbol{x}) > 1 \\ 1 - y\mathbf{w}^\top \Phi(\boldsymbol{x}) - \frac{\tau}{2} & \text{if } y\mathbf{w}^\top \Phi(\boldsymbol{x}) < 1 - \tau \\ \frac{1}{2\tau}\left(1 - y\mathbf{w}^\top \Phi(\boldsymbol{x})\right)^2 & \text{otherwise} \end{cases}$$

$$\nabla_o l(y, o) = -\mathbb{I}_{\{yo < 1 - \tau\}} y + \tau^{-1} \mathbb{I}_{1-\tau \leq yo \leq 1}(yo - 1) y$$
$$|\nabla_o l(y, o)| = \left|\mathbb{I}_{\{yo < 1-\tau\}}\right| + \left|\tau^{-1} \mathbb{I}_{1-\tau \leq yo \leq 1}(yo - 1)\right|$$
$$\leq \left|\mathbb{I}_{\{yo < 1-\tau\}}\right| + \tau^{-1}\tau \left|\mathbb{I}_{1-\tau \leq yo \leq 1}\right| \leq 1 = A$$

**Acknowledgment**: This work is partially supported by the Australian Research Council under the Discovery Project DP160109394.

**$\ell_1$ loss**

$$l(y, o) = |y - o|$$

$$l(\mathbf{w}, \boldsymbol{x}, y) = |y - \mathbf{w}^\top \Phi(\boldsymbol{x})|$$

$$\nabla_o l(y, o) = \mathbf{sign}(o - y)$$

$$|\nabla_o l(y, o)| \leq 1 = A$$

**$\varepsilon$-insensitive loss**

$$l(y, o) = \max(0, |y - o| - \varepsilon)$$

$$l(\mathbf{w}, \boldsymbol{x}, y) = \max\left(0, |y - \mathbf{w}^\top \Phi(\boldsymbol{x})| - \varepsilon\right)$$

$$\nabla_o l(y, o) = \mathbb{I}_{|y-o| \geq \varepsilon} \mathbf{sign}(o - y)$$

$$|\nabla_o l(y, o)| \leq 1 = A$$

We note that $\mathbb{I}_A$ denotes the indicator function which renders 1 if $A$ is true and 0 otherwise.

## 2  Proofs

**Lemma 1.** *After the iteration $t$, we have the following representations*

$$\hat{\mathbf{w}}_t = \sum_{j=1}^{t} \alpha_j (1 - \beta_j) \Phi(\boldsymbol{x}_j) \tag{1}$$

$$\tilde{\mathbf{w}}_t = \sum_{j=1}^{t} \alpha_j \beta_j \boldsymbol{z}(\boldsymbol{x}_j) \tag{2}$$

$$\mathbf{w}_t = \sum_{j=1}^{t} \alpha_j \Phi(\boldsymbol{x}_j) \tag{3}$$

*where $\alpha_j = -\eta_t \nabla_o l\left(y_j, f_j^h(\boldsymbol{x}_j)\right), \forall j = 1, \ldots, t$ and $\eta_t = \frac{1}{\lambda t}$.*

*Proof.* Since if $\beta_j = 1$, we perform the budget maintenance procedure and move the current vector to the random-feature space, we have the representations in Eqs. (1,2,3). In addition at the iteration $j$, $\Phi(\boldsymbol{x}_j)$ arrives with the initial coefficient $\alpha_j = -\eta_j \nabla_o l\left(y_j, f_j^h(\boldsymbol{x}_j)\right)$. After the iteration $t > j$, this coefficient becomes

$$\alpha_j = -\frac{t-1}{t}\frac{t-2}{t-1}\cdots\frac{j}{j+1}\frac{1}{\lambda j}\nabla_o l\left(y_i, f_j^h(\boldsymbol{x}_j)\right) = -\eta_t \nabla_o l\left(y_j, f_j^h(\boldsymbol{x}_j)\right)$$

$\square$

**Theorem 2.** *With a probability at least $1 - 2^8 \left(\frac{\sigma_\mu A d_\mathcal{X}}{\lambda \varepsilon}\right) \exp\left(-\frac{D\lambda^2 \varepsilon^2}{4(M+2)A^2}\right)$ where $d_\mathcal{X}$ specifies the diameter of the compact set $\mathcal{X}$, we have*

*i) $\left|f_t(\boldsymbol{x}) - f_t^h(\boldsymbol{x})\right| \leq \varepsilon$ for all $t > 0$ and $\boldsymbol{x} \in \mathcal{X}$.*

*ii) $\mathbb{E}\left[\left|f_t(\boldsymbol{x}) - f_t^h(\boldsymbol{x})\right|\right] \leq A^{-1}\lambda\varepsilon \sum_{j=1}^{t} \mathbb{E}\left[\alpha_j^2\right]^{1/2} \mu_j^{1/2}$ where $\mu_j = p(\beta_j = 1)$.*

Let us define a random map $z : \mathbb{R}^d \to \mathbb{R}^{2D}$ where $z(x) = \frac{1}{D^{1/2}}\left[\cos\left(\omega_i^\top x\right), \sin\left(\omega_i^\top x\right)\right]_{i=1}^{D}$ and $\omega_1, \ldots, \omega_D \overset{i.i.d}{\sim} \mathcal{N}\left(0, \sigma^{-2}I\right)$ for every $x \in \mathbb{R}^d$. We would like to restate Claim 1 in [3].

Let $\mathcal{M}$ be a compact subset of $\mathbb{R}^d$ with diameter $diam(\mathcal{M})$. Then, for the random mapping $z(.)$, we have

$$\mathbb{P}\left(\sup_{x,x' \in \mathcal{M}} \left|K\left(x, x'\right) - z(x)^\top z\left(x'\right)\right| < \varepsilon\right) \geq 1 - 2^8 \left(\frac{\sigma \, diam(\mathcal{M})}{\varepsilon}\right) exp\left(\frac{-D\varepsilon^2}{4(d+2)}\right)$$

where $K\left(x, x^{'}\right) = e^{-\frac{\left\|x - x^{'}\right\|^2}{2\sigma^2}}$.

*Proof.* We denote

$$\omega = (\omega_1, ..., \omega_D) \sim p_\omega(\omega) = \prod_{i=1}^{D} \mathcal{N}\left(\omega_i | 0, \sigma^{-2}I\right)$$

$$\tilde{K}\left(x, x^{'}\right) = z(x)^{\mathsf{T}} z\left(x^{'}\right) = D^{-1} \sum_{i=1}^{D} \left(cos\left(\omega_i^{\mathsf{T}} x\right) cos\left(\omega_i^{\mathsf{T}} x^{'}\right) + sin\left(\omega_i^{\mathsf{T}} x\right) sin\left(\omega_i^{\mathsf{T}} x^{'}\right)\right)$$

We further denote

$$g(\omega) = \sup_{x,x^{'} \in \mathcal{M}} \left|K\left(x, x^{'}\right) - \tilde{K}\left(x, x^{'}\right)\right|$$

$$G_\varepsilon = \left\{\omega : g(\omega) < A^{-1}\lambda\varepsilon\right\}$$

It is certain that $\mathbb{P}_\omega(G_\varepsilon) \geq 1 - \theta$ where $\theta = 2^8 \left(\frac{\sigma A \operatorname{diam}(\mathcal{M})}{\lambda\varepsilon}\right) exp\left(\frac{-D\lambda^2\varepsilon^2}{4(d+2)A^2}\right)$ and for every $\omega \in G_\varepsilon$ and $x, x^{'} \in \mathcal{M}$ we have

$$\left|K\left(x, x^{'}\right) - \tilde{K}\left(x, x^{'}\right)\right| < A^{-1}\lambda\varepsilon$$

We now turn back to Theorem 2. It appears that

$$\left|f_t(x) - f_t^h(x)\right| \leq \sum_{j=1}^{t} \beta_j |\alpha_j| \left|K(x_j, x) - \tilde{K}(x_j, x)\right|$$

Therefore, for every $\omega \in G_\varepsilon$ we have

$$\left|f_t(x) - f_t^h(x)\right| \leq A^{-1}\lambda\varepsilon \sum_{j=1}^{t} \beta_j |\alpha_j|$$

Let us denote $s = (x_1, y_1), ..., (x_t, y_t)$. Taking expectation of the above inequality w.r.t $s$, we gain for all $\omega \in G_\varepsilon$

$$\mathbb{E}_s\left[\left|f_t(x) - f_t^h(x)\right|\right] \leq A^{-1}\lambda\varepsilon \sum_{j=1}^{t} \mathbb{E}_s\left[\beta_j^2\right]^{1/2} \mathbb{E}_s\left[\alpha_j^2\right]^{1/2}$$

$$\leq A^{-1}\lambda\varepsilon \sum_{j=1}^{t} \mu_j \mathbb{E}_s\left[\alpha_j^2\right]^{1/2}$$

It means that

$$\mathbb{P}_\omega\left(\mathbb{E}_s\left[\left|f_t(x) - f_t^h(x)\right|\right] \leq A^{-1}\lambda\varepsilon \sum_{j=1}^{t} \mu_j \mathbb{E}_s\left[\alpha_j^2\right]^{1/2}\right) \geq \mathbb{P}_\omega(G_\varepsilon) \geq 1 - \theta$$

$\square$

**Lemma 3.** *The following statement holds for all $t$*

$$\|\mathbf{w}_t\| \leq \frac{A}{\lambda}$$

*Proof.* Using Lemma 1, we have

$$\mathbf{w}_t = \sum_{j=1}^{t} \alpha_j \Phi\left(\boldsymbol{x}_j\right)$$

where $\alpha_j = -\eta_t \nabla_o l\left(y_j, f_j^h\left(\boldsymbol{x}_j\right)\right)$.

It implies that

$$\|\mathbf{w}_t\| \leq \sum_{j=1}^{t} |\alpha_j| \|\Phi\left(\boldsymbol{x}_j\right)\| \leq \sum_{j=1}^{t} |\alpha_j| \leq \sum_{j=1}^{t} \frac{A}{\lambda t} = \frac{A}{\lambda}$$

□

**Lemma 4.** *The following statement holds for all* $t$

$$\|g_t\| \leq G = 2A$$

*where we define* $g_t = \lambda \mathbf{w}_t + \nabla_{\mathbf{w}} l\left(\mathbf{w}_t, \boldsymbol{x}_t, y_t\right) = \lambda \mathbf{w}_t + \nabla_o l\left(y_t, f_t\left(\boldsymbol{x}_t\right)\right) \Phi\left(\boldsymbol{x}_t\right)$.

*Proof.* We derive as

$$\|g_t\| \leq \lambda \|\mathbf{w}_t\| + \|\nabla_o l\left(y_t, f_t\left(\boldsymbol{x}_t\right)\right) \Phi\left(\boldsymbol{x}_t\right)\| \leq \lambda \frac{A}{\lambda} + A = 2A$$

□

**Lemma 5.** *The following statement holds for all* $t$

$$\mathbb{E}\left[\|\mathbf{w}_t - \mathbf{w}^\star\|^2\right] \leq W^2$$

*where* $W = \frac{2A\left(1+\sqrt{5}\right)}{\lambda}$.

*Proof.* Recall that $g_t = \lambda \mathbf{w}_t + \nabla_{\mathbf{w}} l\left(\mathbf{w}_t, \boldsymbol{x}_t, y_t\right) = \lambda \mathbf{w}_t + \nabla_o l\left(y_t, f_t\left(\boldsymbol{x}_t\right)\right) \Phi\left(\boldsymbol{x}_t\right)$. It is obvious that $g_t$ satisfies

$$\mathbb{E}_{\left(\boldsymbol{x}_t, y_t\right)}\left[g_t | \mathbf{w}_t\right] = \mathcal{J}^{'}\left(\mathbf{w}_t\right)$$

We have the following if we denote $\delta g_t = g_t - g_t^h$

$$\|\mathbf{w}_{t+1} - \mathbf{w}^\star\|^2 = \|\mathbf{w}_t - \eta_t g_t^h - \mathbf{w}^\star\| = \|\mathbf{w}_t - \eta_t g_t - \mathbf{w}^\star + \eta_t \delta g_t\|^2$$
$$= \|\mathbf{w}_t - \mathbf{w}^\star\|^2 - 2\eta_t g_t^\top\left(\mathbf{w}_t - \mathbf{w}^\star\right) + \eta_t^2 \|g_t\|^2 - 2\eta_t^2 g_t^\top \delta g_t + \eta_t^2 \|\delta g_t\|^2 + 2\eta_t\left(\mathbf{w}_t - \mathbf{w}^\star\right)^\top \delta g_t$$

It appears that

$$\delta g_t = \left[\nabla_o l\left(y_t, f_t\left(\boldsymbol{x}_t\right)\right) - \nabla_o l\left(y_t, f_t^h\left(\boldsymbol{x}_t\right)\right)\right] \Phi\left(\boldsymbol{x}_t\right)$$
$$\|\delta g_t\| = \left|\nabla_o l\left(y_t, f_t\left(\boldsymbol{x}_t\right)\right) - \nabla_o l\left(y_t, f_t^h\left(\boldsymbol{x}_t\right)\right)\right| \leq 2A$$

Hence, we obtain

$$\|\mathbf{w}_{t+1} - \mathbf{w}^\star\|^2 \leq \|\mathbf{w}_t - \mathbf{w}^\star\|^2 - 2\eta_t g_t^\top\left(\mathbf{w}_t - \mathbf{w}^\star\right) + \eta_t^2 G^2 + 4\eta_t^2 GA + 4\eta_t^2 A^2$$
$$+ 2\eta_t \|\mathbf{w}_t - \mathbf{w}^\star\| \|\delta g_t\|$$

Taking conditional expectation w.r.t $\mathbf{w}_t$ on both sides of the above inequality, we gain

$$\mathbb{E}\left[\|\mathbf{w}_{t+1} - \mathbf{w}^\star\|^2\right] \leq \mathbb{E}\left[\|\mathbf{w}_t - \mathbf{w}^\star\|^2\right] - 2\eta_t \nabla_{\mathbf{w}} \mathcal{J}\left(\mathbf{w}_t\right)^\top\left(\mathbf{w}_t - \mathbf{w}^\star\right) + \eta_t^2 G^2 + 4\eta_t^2 GA$$
$$+ 4\eta_t^2 A^2 + 2\eta_t \mathbb{E}\left[\|\mathbf{w}_t - \mathbf{w}^\star\| \|\delta g_t\|\right]$$
$$\leq \mathbb{E}\left[\|\mathbf{w}_t - \mathbf{w}^\star\|^2\right] + 16A^2 \eta_t^2 + 2\eta_t \mathbb{E}\left[\|\mathbf{w}_t - \mathbf{w}^\star\| \|\delta g_t\|\right] - \frac{1}{t} \|\mathbf{w}_t - \mathbf{w}^\star\|$$

Here we note that we have used

$$\nabla_{\mathbf{w}} \mathcal{J}\left(\mathbf{w}_t\right)^\top\left(\mathbf{w}_t - \mathbf{w}^\star\right) \geq \mathcal{J}\left(\mathbf{w}_t\right) - \mathcal{J}\left(\mathbf{w}^\star\right) + \frac{\lambda}{2} \|\mathbf{w}_t - \mathbf{w}^\star\|^2 \geq \frac{\lambda}{2} \|\mathbf{w}_t - \mathbf{w}^\star\|^2$$

Taking expectation on both sides again, we obtain

$$\mathbb{E}\left[\|\mathbf{w}_{t+1} - \mathbf{w}^\star\|^2\right] \leq \frac{t-1}{t}\mathbb{E}\left[\|\mathbf{w}_t - \mathbf{w}^\star\|^2\right] + \frac{16A^2}{\lambda^2 t^2} + \frac{4A\mathbb{E}\left[\|\mathbf{w}_t - \mathbf{w}^\star\|^2\right]^{1/2}}{\lambda t}$$

$$\leq \frac{t-1}{t}\mathbb{E}\left[\|\mathbf{w}_t - \mathbf{w}^\star\|^2\right] + \frac{16A^2}{\lambda^2 t} + \frac{4A\mathbb{E}\left[\|\mathbf{w}_t - \mathbf{w}^\star\|^2\right]^{1/2}}{\lambda t}$$

Choose $W = \frac{2A(1+\sqrt{5})}{\lambda}$, we have if $\mathbb{E}\left[\|\mathbf{w}_t - \mathbf{w}^\star\|^2\right] \leq W^2$ then $\mathbb{E}\left[\|\mathbf{w}_{t+1} - \mathbf{w}^\star\|^2\right] \leq W^2$. $\square$

**Theorem 6.** *The following statement guarantees for all $T$*

$$\mathbb{E}\left[\mathcal{J}\left(\overline{\mathbf{w}}_T\right) - \mathcal{J}\left(\mathbf{w}^\star\right)\right] \leq \mathbb{E}\left[\frac{1}{T}\sum_{t=1}^{T}\mathcal{J}\left(\mathbf{w}_t\right) - \mathcal{J}\left(\mathbf{w}^\star\right)\right] \leq \frac{8A^2\left(\log\left(T\right)+1\right)}{\lambda T} + \frac{1}{T}W\sum_{t=1}^{T}\mathbb{E}\left[M_t^2\right]^{1/2}$$

*where $\overline{\mathbf{w}}_T = \frac{1}{T}\sum_{t=1}^{T}\mathbf{w}_t$, $M_t = \nabla_o l\left(y_t, f_t\left(\boldsymbol{x}_t\right)\right) - \nabla_o l\left(y_t, f_t^h\left(\boldsymbol{x}_t\right)\right)$.*

*Proof.* Recall that $g_t = \lambda\mathbf{w}_t + \nabla_\mathbf{w} l\left(\mathbf{w}_t, \boldsymbol{x}_t, y_t\right) = \lambda\mathbf{w}_t + \nabla_o l\left(y_t, f_t\left(\boldsymbol{x}_t\right)\right)\Phi\left(\boldsymbol{x}_t\right)$. It is obvious that $g_t$ satisfies

$$\mathbb{E}_{(\boldsymbol{x}_t, y_t)}\left[g_t | \mathbf{w}_t\right] = \nabla_\mathbf{w}\mathcal{J}\left(\mathbf{w}_t\right)$$

We have the following if we denote $\delta g_t = g_t - g_t^h$

$$\|\mathbf{w}_{t+1} - \mathbf{w}^\star\|^2 = \|\mathbf{w}_t - \eta_t g_t^h - \mathbf{w}^\star\| = \|\mathbf{w}_t - \eta_t g_t - \mathbf{w}^\star + \eta_t \delta g_t\|^2$$
$$= \|\mathbf{w}_t - \mathbf{w}^\star\|^2 - 2\eta_t g_t^\top\left(\mathbf{w}_t - \mathbf{w}^\star\right) + \eta_t^2 \|g_t\|^2 - 2\eta_t^2 g_t^\top \delta g_t + \eta_t^2 \|\delta g_t\|^2 + 2\eta_t\left(\mathbf{w}_t - \mathbf{w}^\star\right)^\top \delta g_t$$

It appears that

$$\delta g_t = \left[\nabla_o l\left(y_t, f_t\left(\boldsymbol{x}_t\right)\right) - \nabla_o l\left(y_t, f_t^h\left(\boldsymbol{x}_t\right)\right)\right]\Phi\left(x_t\right)$$
$$\|\delta g_t\| = \left|\nabla_o l\left(y_t, f_t\left(\boldsymbol{x}_t\right)\right) - \nabla_o l\left(y_t, f_t^h\left(\boldsymbol{x}_t\right)\right)\right| \leq 2A$$

Hence, we obtain

$$\|\mathbf{w}_{t+1} - \mathbf{w}^\star\|^2 \leq \|\mathbf{w}_t - \mathbf{w}^\star\|^2 - 2\eta_t g_t^\top\left(\mathbf{w}_t - \mathbf{w}^\star\right) + \eta_t^2 G^2 + 4\eta_t^2 GA + 4\eta_t^2 A^2$$
$$+ 2\eta_t \|\mathbf{w}_t - \mathbf{w}^\star\| \|\delta g_t\|$$

$$g_t^\top\left(\mathbf{w}_t - \mathbf{w}^\star\right) \leq \frac{\|\mathbf{w}_t - \mathbf{w}^\star\|^2 - \|\mathbf{w}_{t+1} - \mathbf{w}^\star\|^2}{2\eta_t} + 8A^2\eta_t + \|\mathbf{w}_t - \mathbf{w}^\star\| \|\delta g_t\|$$

Taking conditional expectation w.r.t $\mathbf{w}_t$ on both sides, we gain

$$\nabla_\mathbf{w}\mathcal{J}\left(\mathbf{w}_t\right)^\top\left(\mathbf{w}_t - \mathbf{w}^\star\right) \leq \mathbb{E}\left[\frac{\|\mathbf{w}_t - \mathbf{w}^\star\|^2}{2\eta_t}\right] - \mathbb{E}\left[\frac{\|\mathbf{w}_{t+1} - \mathbf{w}^\star\|^2}{2\eta_t}\right] + 8A^2\eta_t + \mathbb{E}\left[\|\mathbf{w}_t - \mathbf{w}^\star\| \|\delta g_t\|\right]$$

$$\mathcal{J}\left(\mathbf{w}_t\right) - \mathcal{J}\left(\mathbf{w}^\star\right) + \frac{\lambda}{2}\|\mathbf{w}_t - \mathbf{w}^\star\|^2 \leq \mathbb{E}\left[\frac{\|\mathbf{w}_t - \mathbf{w}^\star\|^2}{2\eta_t}\right] - \mathbb{E}\left[\frac{\|\mathbf{w}_{t+1} - \mathbf{w}^\star\|^2}{2\eta_t}\right]$$
$$+ 8A^2\eta_t + \mathbb{E}\left[\|\mathbf{w}_t - \mathbf{w}^\star\| \|\delta g_t\|\right]$$

Taking expectation on both sides once again, we achieve

$$\mathbb{E}\left[\mathcal{J}\left(\mathbf{w}_t\right) - \mathcal{J}\left(\mathbf{w}^\star\right)\right] \leq \frac{\lambda}{2}\left(t-1\right)\mathbb{E}\left[\|\mathbf{w}_t - \mathbf{w}^\star\|^2\right] - \frac{\lambda}{2}t\mathbb{E}\left[\|\mathbf{w}_{t+1} - \mathbf{w}^\star\|^2\right]$$
$$+ 8A^2\eta_t + \mathbb{E}\left[\|\mathbf{w}_t - \mathbf{w}^\star\| \|\delta g_t\|\right]$$

$$\mathbb{E}\left[\mathcal{J}\left(\mathbf{w}_t\right) - \mathcal{J}\left(\mathbf{w}^\star\right)\right] \leq \frac{\lambda}{2}\left(t-1\right)\mathbb{E}\left[\|\mathbf{w}_t - \mathbf{w}^\star\|^2\right] - \frac{\lambda}{2}t\mathbb{E}\left[\|\mathbf{w}_{t+1} - \mathbf{w}^\star\|^2\right]$$
$$+ 8A^2\eta_t + \mathbb{E}\left[\|\mathbf{w}_t - \mathbf{w}^\star\|^2\right]^{1/2}\mathbb{E}\left[\|\delta g_t\|^2\right]^{1/2}$$

Taking sum the above inequality when $t = 1, ..., T$, we obtain

$$\mathbb{E}\left[\frac{1}{T}\sum_{t=1}^{T}\mathcal{J}\left(\mathbf{w}_t\right) - \mathcal{J}\left(\mathbf{w}^\star\right)\right] \leq \frac{8A^2}{\lambda}\sum_{t=1}^{T}\frac{1}{t} + \frac{1}{T}W\sum_{t=1}^{T}\mathbb{E}\left[M_t^2\right]^{1/2}$$

$$\leq \frac{8A^2\left(\log T + 1\right)}{\lambda T} + \frac{1}{T}W\sum_{t=1}^{T}\mathbb{E}\left[M_t^2\right]^{1/2}$$

Here we note that

$$\|\delta g_t\| = \left\|\left[\nabla_o l\left(y_t, f_t\left(\boldsymbol{x}_t\right)\right) - \nabla_o l\left(y_t, f_t^h\left(\boldsymbol{x}_t\right)\right)\right]\Phi\left(\boldsymbol{x}_t\right)\right\| = |M_t|$$

The last conclusion comes from the convexity of the function $\mathcal{J}\left(.\right)$.  $\square$

**Theorem 7.** *Assume that $l\left(y, o\right)$ is a $\gamma$-strongly smooth loss function. With a probability at least $1 - \theta$, the following statements hold*

*i)* $\quad\mathbb{E}\left[\mathcal{J}\left(\overline{\mathbf{w}}_T\right) - \mathcal{J}\left(\mathbf{w}^\star\right)\right] \quad \leq \quad \mathbb{E}\left[\frac{1}{T}\sum_{t=1}^{T}\mathcal{J}\left(\mathbf{w}_t\right) - \mathcal{J}\left(\mathbf{w}^\star\right)\right] \quad \leq \quad \frac{8A^2\left(\log T+1\right)}{\lambda T} \quad +$
$\frac{1}{T}W\gamma\varepsilon\sum_{t=1}^{T}\left(\frac{\sum_{i=1}^{t}\mu_i}{t}\right)^{1/2}$

*ii)* $\mathbb{E}\left[\mathcal{J}\left(\overline{\mathbf{w}}_T\right) - \mathcal{J}\left(\mathbf{w}^\star\right)\right] \leq \mathbb{E}\left[\frac{1}{T}\sum_{t=1}^{T}\mathcal{J}\left(\mathbf{w}_t\right) - \mathcal{J}\left(\mathbf{w}^\star\right)\right] \leq \frac{8A^2\left(\log T+1\right)}{\lambda T} + W\gamma\varepsilon$

*where* $\theta = 2^8\left(\frac{\sigma_\mu A d_\mathcal{X}}{\lambda\varepsilon}\right)\exp\left(-\frac{D\lambda^2\varepsilon^2}{4\left(M+2\right)A^2}\right).$

*Proof.* From the smoothness of the loss function, we have

$$\left|\nabla_o l\left(y_t, f_t\left(\boldsymbol{x}_t\right)\right) - \nabla_o l\left(y_t, f_t^h\left(\boldsymbol{x}_t\right)\right)\right| \leq \gamma\left|f_t\left(\boldsymbol{x}_t\right) - f_t^h\left(\boldsymbol{x}_t\right)\right|$$

Referring to Lemma 2, with a probability at least $1 - 2^8\left(\frac{\sigma_\mu A d_\mathcal{X}}{\lambda\varepsilon}\right)\exp\left(-\frac{D\lambda^2\varepsilon^2}{4\left(M+2\right)A^2}\right) = 1 - \theta$ we have

$$|M_t| \leq \gamma A^{-1}\lambda\varepsilon\sum_{j=1}^{t}\left|\alpha_j\right|\beta_i \leq \gamma A^{-1}\lambda\varepsilon\sum_{j=1}^{t}\frac{A}{\lambda t}\beta_j = \frac{\gamma\varepsilon}{t}\sum_{j=1}^{t}\beta_j$$

$$M_t^2 \leq \frac{\gamma^2\varepsilon^2}{t^2}\left(\sum_{j=1}^{t}\beta_j\right)^2 \leq \frac{\gamma^2\varepsilon^2}{t}\sum_{j=1}^{t}\beta_j^2 = \frac{\gamma^2\varepsilon^2}{t}\sum_{j=1}^{t}\beta_j \quad \text{(since } \beta_i = 0 \text{ or } 1\text{)}$$

$$\mathbb{E}\left[M_t^2\right] \leq \frac{\gamma^2\varepsilon^2}{t}\left(\sum_{j=1}^{t}\mu_j\right)$$

and $|M_t| \leq \gamma\varepsilon$. Therefore, with a probability at least $1 - \theta$ we achieve

$$\mathbb{E}\left[\mathcal{J}\left(\overline{\mathbf{w}}_T\right) - \mathcal{J}\left(\mathbf{w}^\star\right)\right] \leq \mathbb{E}\left[\frac{1}{T}\sum_{t=1}^{T}\mathcal{J}\left(\mathbf{w}_t\right) - \mathcal{J}\left(\mathbf{w}^\star\right)\right]$$

$$\leq \frac{8A^2\left(\log T+1\right)}{\lambda T} + \frac{1}{T}W\gamma\varepsilon\sum_{t=1}^{T}\frac{\left(\sum_{j=1}^{t}\mu_j\right)^{1/2}}{t^{1/2}}$$

$$\leq \frac{8A^2\left(\log T+1\right)}{\lambda T} + \frac{1}{T}W\gamma\varepsilon\sum_{t=1}^{T}\left(\frac{\sum_{j=1}^{t}\mu_j}{t}\right)^{1/2}$$

and

$$\mathbb{E}\left[\mathcal{J}\left(\overline{\mathbf{w}}_T\right) - \mathcal{J}\left(\mathbf{w}^\star\right)\right] \leq \mathbb{E}\left[\frac{1}{T}\sum_{t=1}^{T}\mathcal{J}\left(\mathbf{w}_t\right) - \mathcal{J}\left(\mathbf{w}^\star\right)\right]$$

$$\leq \frac{8A^2\left(\log T + 1\right)}{\lambda T} + \frac{1}{T}W\sum_{t=1}^{T}\gamma\varepsilon$$

$$\leq \frac{8A^2\left(\log T + 1\right)}{\lambda T} + W\gamma\varepsilon$$

$\square$

## 3 Computational Complexities of DualSGD and FOGD

We compare the computational complexities of our proposed DualSGD and Fourier Online Gradient Descent (FOGD) [2]. Recall that $M$ and $D$ denote the dimensions of input space and feature space, and $B$ the budget size. There are four operators: (i) random feature mapping; (ii) kernel function; (iii) sorting coefficients of support vectors and (iv) prediction. The random feature mapping first projects the input data vector to random feature space with $\mathcal{O}(MD)$ computational complexity, and then compute *sin, cos* on the random feature dimension with $\mathcal{O}\left(2D * 2^{\log n}n\log^2 n\right)$ where $n$ is the number of bits accuracy [1]. The kernel function, sorting coefficients and prediction operate in $\mathcal{O}(MB)$, $\mathcal{O}(B\log B)$ and $\mathcal{O}(D)$ complexity, respectively. The FOGD performs random feature mapping and prediction whilst the DualSGD performs all four operators.

Let $D_1$ and $D_2$ denote the number of random features of FOGD and DualSGD. The computational complexities of FOGD and DualSGD reads

$$\mathcal{O}_{\text{FOGD}} = \mathcal{O}\left(MD_1 + 2D_1 * 2^{\log n}n\log^2 n + D_1\right) = U\left(MD_1 + 2D_1 * 2^{\log n}n\log^2 n + D_1\right)$$

$$\mathcal{O}_{\text{DualSGD}} = \mathcal{O}\left(MD_2 + 2D_2 * 2^{\log n}n\log^2 n + D_2 + MB + B\log B\right)$$

$$= V\left(MD_2 + 2D_2 * 2^{\log n}n\log^2 n + D_2 + MB + B\log B\right)$$

where $U, V$ are the number of iterations.

Taking the subtraction of $\mathcal{O}_{\text{FOGD}}$ and $\mathcal{O}_{\text{DualSGD}}$, we obtain:

$$\hat{\mathcal{O}} = \mathcal{O}_{\text{FOGD}} - \mathcal{O}_{\text{DualSGD}}$$

$$= M\left(UD_1 - VD_2 - B\right) + \left(UD_1 - VD_2\right)\left(2 * 2^{\log n}n\log^2 n + 1\right) - B\log B$$

According Fig. 1 in the introduction section, $D_1 \gg D_2$ and $D_1 \gg B$, thus $D_1 - D_2 \gg B$. In addition, we assume that $U = V$ and normally use double-precision floating-point with $n = 64$ (bits) for storing and computing real number, thus $2 * 2^{\log n}n\log^2 n + 1 > \log B$. Finally, we can see that $\mathcal{O} \gg 0$, thus the computational complexity of DualSGD, in practice, is significantly lower than that of FOGD.