[Reviews · NeurIPS 2016]

Reviewer 1

Summary

The paper introduces a new algorithm for kernel-based online learning, where (a) the number of support vectors is controlled by discarding the least significant ones, and (b) information of the discarded support vectors is partially retained by maintaining a random projection of them in a lower-dimensional space. Regret bounds are derived in terms of the "budget" of retained support vectors and the dimensionality of the random projections. Experiments show that compared to previous methods, the new algorithm obtains a very good trade-off between prediction accuracy and memory use.

Qualitative Assessment

The rebuttal answered all my technical questions, in particular the one about probability and expectations in Theorems 1(ii) and 3. My scores (which were given assuming the technicalities can be fixed) remain the same. *** Original review below *** The algorithm is as far as I know an original and intuitively appealing combination of techniques previously introduced in this quite active subarea. I quite like it. The theoretical results do not feel that exciting. The regret bounds seem more or less what one might expect, and there is no comparison to bounds for other algorithms. I understand that there are no directly comparable bounds since you have both budget size and dimensionality of random projections as significant parameters, but still some discussion would be nice. The dependence on budget B in your bounds could perhaps be pointed out more explicitly. The experiments are compelling. The relevant competing algorithms are included, and the data sets are sufficiently varied. Detailed comments: Algorithm 1: On line 4, you include all indices in I_t, even when alpha_t is zero. Is this really intended? On the other hand, b is increased only for non-zero alpha_t. You also fail to mention where b is decreased, but it would seem if you really do this as written then the arg min in Algorithm 2 might just catch a lot of zeros and not actually free up any budget. Theorem 1: I did not find a definition for sigma_mu in [15]. There was sigma_P where P is a distribution. But what is mu here, anyway? Theorem 2: "the following statement guarantees" seems wrong line 169: "where" should be "were"? Experiments: Is it reasonable to use the same values for hyperparameters such as B for all the algorithms? Would it make any difference in the results if each algorithm was individually tuned? line 232: Pegasos is not mentioned anywhere else as far as I can see line 257, "due to additional exponential operators": Another factor that causes more computation in Logit than in Hinge is that the gradient is always non-zero. reference [15]: "Infomration" Appendix, proof of Theorem 2: To get to (4) you need the previous bound to hold for all j, but if I understand correctly the current bound is just for a fixed j. Proof of Theorem 2, last chain of inequalities: To get here you seem to need E[alpha^2 beta^2] \leq E[alpha^2] E[beta^2] , which seems intutively reasonable but would require a proof. Similar issue appear in proof of Theorem 6 (top of page 5). page 5, lines 7 and 8: The term with 8A^2 seems to have a factor 1/lambda missing. Proof of Theorem 7: There is no "Lemma 2" here. Appendix, Section 3: Your use of O notation is a bit sloppy. In particular, from f_1 = O(g_1) and f_2 = O(g_2) one cannot conclude f_1-f_2 = O(g_1-g_2).

Confidence in this Review

2-Confident (read it all; understood it all reasonably well)


Reviewer 2

Summary

This paper presents a new stochastic gradient method for handling curse of kernelization. More specifically, authors introduce the provision vector which includes the information of removed vectors from a subset of data points. Based on the precision vector, author proposed a new budget maintenance strategy named k-merging, and showed the convergence rate of the proposed algorithm. Finally, through experiments, authors compared the proposed method with several existing state-of-the-art methods and showed that the proposed algorithm compares favorably with existing methods.

Qualitative Assessment

This paper presents a new stochastic gradient method for handling curse of kernelization. More specifically, authors introduce the provision vector which includes the information of removed vectors from a subset of data points. Based on the precision vector, author proposed a new budget maintenance strategy named k-merging, and showed the convergence rate of the proposed algorithm. Finally, through experiments, authors compared the proposed method with several existing state-of-the-art methods and showed that the proposed algorithm compares favorably with existing methods. The paper is clearly written and easy to follow. Overall, the proposed approach is simple yet effective, and has theoretical guarantee. Thus, I tends to vote for acceptance. Detailed comments: 1. From the theorem1 and 2, is it possible to theoretically show in which case your proposed method outperforms compared to existing algorithms? Additional interpretation of the theorem would be helpful. 2. In the regression model, why you use two different mapping functions (Phi(x) and z(x))? Is it possible to use random projections for both of them? 3. In experimental section, why the mistake rates of methods are quite different? Since the objective function is convex, I wonder where the difference comes from. Perhaps, basis selection? Or cross validation? I think it would be good to have comparisons with fixed kernel parameter. 4. For those who are not familiar with Budgeted learning, it is hard to understand the meaning for removing. The explanation of budgeted learning in Section2 would be helpful for non-budgeted learning researchers.

Confidence in this Review

2-Confident (read it all; understood it all reasonably well)


Reviewer 3

Summary

The paper gives an algorithm for doing some form of online learning involving kernels in which there is some kind of budget.

Qualitative Assessment

I found the presentation of this paper so poor, that I really could not even tell what the problem is that is being solved, or what the proposed algorithm is. The abstract is very hard to penetrate, and I kept reading, hoping that things would be explained better, but that never seemed to happen. Key terms like budget, removal, projection, merging are used repeatedly without ever being explained, as far as I can tell. The notion of a "random feature space" is never explained. The main ideas of the algorithm in section 2.3 are given very little or no motivation. Even the problem setting is not well explained. The paper claims initially to be about online learning, but the problem given in (1) seems to be about a batch setting. But then the paper switches back to an online setting, sort of, in the experiments. The experimental results are possibly impressive, if they could be understood. The experiments seem to be thoroughly explained, although what comes across is what a lot of parameters the new method has, and what a challenge it is to tune them. A few other comments: Should the expectations be removed in Theorem 3? It seems odd to have both high probability and expectation in the same statement. The notation f_t^h and f_t are introduced at lines 126-127, but are not explained for another half page. Very confusing.

Confidence in this Review

1-Less confident (might not have understood significant parts)


Reviewer 4

Summary

The authors propose a simple dual-space implementation to address the computational complexity involved in employing random features to approximate the kernel function. The proposed method uses an auxiliary space to store information about features that have been discarded. One useful characteristic of the proposed method is that it helps treat in a rather unified manner the three traditional strategies of removal, projection, and merging. There is sufficient analysis in the paper and the results are further illustrated by supporting simulations.

Qualitative Assessment

The authors propose a simple dual-space implementation to address the computational complexity involved in employing random features to approximate the kernel function. The proposed method uses an auxiliary space to store information about features that have been discarded. One useful characteristic of the proposed method is that it helps treat in a rather unified manner the three traditional strategies of removal, projection, and merging. There is sufficient analysis in the paper and the results are further illustrated by supporting simulations.

Confidence in this Review

2-Confident (read it all; understood it all reasonably well)


Reviewer 5

Summary

The paper proposes a Dual Space Gradient Descent (DualSGD) algorithm for online kernel learning. The algorithm balances the input features space and random feature space with ‘k-merging’ technique, thus retaining information and budgeting at the same time. Theoretical analysis shows approximation guarantee. Numerical experiments on online classification and regression demonstrate improved scalability and prediction accuracy.

Qualitative Assessment

Balancing random features and input features to retain information and maintain budge at the same time is an interesting idea. The experiment results look promising ( 30% improved accuracy for the comparable run time) for online classification and regression tasks. But the novelty of the algorithm is relatively limited. It appears to me as learning a kernel with a hybrid of true kernel and random kernel approximation. Thus from algorithmic perspective, I don't see any particular technical difficulty. Also it would be helpful if authors can show the impact of the number of random vectors k. The current theorem relies on the ratio \alpha and \beta. How can we know the values of them in practice?

Confidence in this Review

2-Confident (read it all; understood it all reasonably well)


Reviewer 6

Summary

In this paper, the authors present the Dual Space Gradient Descent, a kernel online learning approach that addresses the "curse of kernelization" by employing a vector in the random-feature space which retains the information of all the vectors that were removed during the budget maintainance process. The authors provide convergence analysis for their algorithm as well as experimental results that show the superiority of their algorithm over the baselines in terms of runtime and predictive performance.

Qualitative Assessment

The paper is well-written and the authors communicate clearly their ideas to the reader. There are some typos (e.g., Line 42: 'require' -> 'required'), however none of them is really major. The problem tackled is potentially of high significance, and the paper seems to contribute towards that goal. Specifically, the proposed algorithm addresses some weaknesses of the state-of-the-art approaches by unifying some existing strategies. The proposed algorithm offers improved performance for both classification and regression tasks and is much more efficient compared to the baselines in terms of the runtime. However, the values of the budget size and the random feature dimension for the proposed algorithm are set to different values compared to these of the baselines. It would be interesting to compare the prediction performance and the runtimes of the various approaches using the same values for these two variables for all methods.

Confidence in this Review

1-Less confident (might not have understood significant parts)